# Inequality in genetic cancer risk suggests bad genes rather than bad luck

Mats Julius Stensrud[1,2] & Morten Valberg[1,3]

Heritability is often estimated by decomposing the variance of a trait into genetic and other factors. Interpreting such variance decompositions, however, is not straightforward. In particular, there is an ongoing debate on the importance of genetic factors in cancer development, even though heritability estimates exist. Here we show that heritability estimates contain information on the distribution of absolute risk due to genetic differences. The approach relies on the assumptions underlying the conventional heritability of liability model. We also suggest a model unrelated to heritability estimates. By applying these strategies, we describe the distribution of absolute genetic risk for 15 common cancers. We highlight the considerable inequality in genetic risk of cancer using different metrics, e.g., the Gini Index and quantile ratios which are frequently used in economics. For all these cancers, the estimated inequality in genetic risk is larger than the inequality in income in the USA.

[1] Oslo Centre for Biostatistics and Epidemiology, Department of Biostatistics, Institute of Basic Medical Sciences, University of Oslo, Postbox 1122 Blindern, 0317 Oslo, Norway. [2] Diakonhjemmet hospital, Department of Medicine, Diakonveien 12, 0370 Oslo, Norway. [3] Oslo Centre for Biostatistics and Epidemiology, Oslo University Hospital, 0370 Oslo, Norway. Correspondence and requests for materials should be addressed to M.J.S. (email: m.j.stensrud@medisin.uio.no)

There are several approaches to quantify the contribution of heritable factors to disease[1,2]. A straightforward strategy is using familial recurrence risks, e.g., the recurrence risk in monozygotic co-twins, given a co-twin is affected ($\lambda_M$), or the recurrence risk in a pair of siblings ($\lambda_S$)[3]. If the relative risk in relatives of affected individuals is different from 1, family related factors influence the risk. Indeed, it has been argued that the majority of such factors are most likely genetic[3–5]. The familial risk estimates may have immediate interest for relatives of affected individuals. These estimates are simple predictors of the individual disease risk, and they may be particularly useful when few other risk factors are known. However, these familial risk estimates per se do not yield accurate information about the magnitude and inequality of genetic and environmental risk[6]. Nor do they indicate the relative importance of heritable, common environmental and other factors. The familial risk estimates are purely observational, and do not have a causal interpretation.

Heritability, on the other hand, allows for comparison between heritable and other factors: The heritability denotes the fraction of the variation of the trait that is due to genetic differences[2]. These estimates are characteristics of the population under study, and cannot be immediately generalised to other populations. To interpret the heritability, we must make assumptions about the underlying causal structure, i.e. we must define a causal model[1,7].

Heritability is often used to evaluate the importance of genetic effects, but the interpretation is not always easy. Intuitively, a large heritability may correspond to a large variability in absolute genetic risk. Nevertheless, it is not straightforward to see how the absolute genetic risk distribution depends on heritability. Indeed, for cancer development the contribution of genetic, environmental factors and chance is debated[8–19, 34, 36], despite the access to heritability data[20].

To better understand the importance of heritable factors, we obtain the distribution of absolute risks due to genetic differences. After estimating the absolute genetic risk distribution, we study the fundamental inequality in cancer risk across individuals, using e.g. Nordic twin data for 15 common cancers[20]. Our analysis suggests that genetic differences lead to substantial inequality in the risk of several cancers.

## Results

**Deriving the distribution of absolute genetic risk**. Human diseases are often considered to be dichotomous traits; you are either affected or unaffected. For such traits, the heritability of liability is frequently used to study inheritance[2]. The concept implies that every individual has a liability to disease, which is the sum of e.g. several genetic and environmental components. Usually the liability is assumed to be normally distributed in the population, and a threshold on the liability scale determines whether an individual acquires the disease. Hence, the standard liability model is usually interpreted as a threshold model[7,21]. This model allows for the decomposition of the variance into genetic and environmental components. It is appealing, because the variance on the liability scale does not depend on the disease prevalence. Furthermore, the normally distributed liability may have some justification in the central limit theorem; if we believe that the liability of a trait is due to several additive genetic and environmental factors, the liability may approximately follow a normal distribution.

In the 1970s a mathematically equivalent interpretation of the threshold model was described, which is based on the genetic liability $\iota_G$, i.e., the liability solely due to genotype[22]. In the Methods section, we have derived the risk of disease given $\iota_G$, which we denote $Y$. Indeed, we express the distribution of $Y$ to study how the genetic risk varies on an individual level.

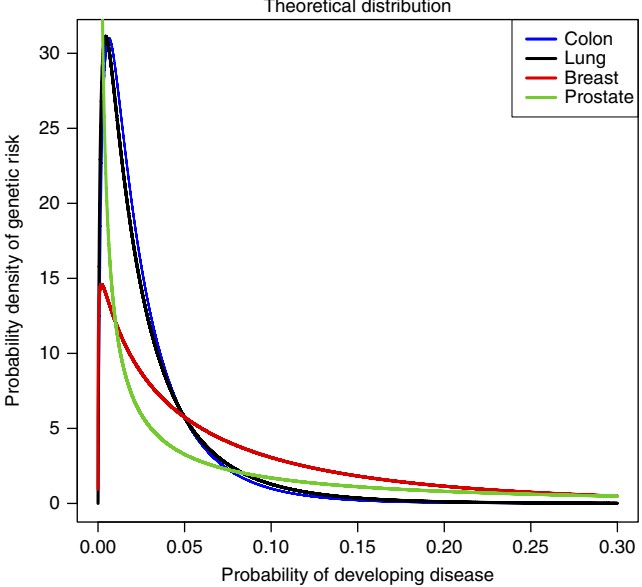

**Fig. 1** Genetic risk distribution for four common cancers. The distribution of risk due to genetic differences is displayed for four common cancers, using heritability and prevalence data from Nordic twin registries[20]

Wray et al.[23] use some similar concepts to see that the probit model fits with real, observed family data[24]. Here, we will use summary estimates of the heritability $h^2$ from twin studies to derive the distribution of $Y$ for 15 common cancers. When the absolute risk distribution is derived, we can obtain various measures of the genetic inequality in risk.

**Exploring inequality in risk for 15 cancers**. Mucci et al.[20] recently reported heritability estimates for 15 common cancers based on the heritability of liability model, using data from Nordic twin registries. We will apply the sampling algorithm described in the Methods section to derive the distribution of absolute risk for these 15 cancers. To illustrate this, Fig. 1 shows the estimated genetic risk distribution for the 4 most common cancers. We interpret the genetic risk as the individual life-time risk of disease, given that the individual's genetic make-up was known, but the environmental exposure unknown. The interpretation relies on the assumptions underlying the heritability of liability model, e.g. that genetic factors and the environmental factors are independent on the liability scale.

By obtaining the risk distributions, we are able to explore the genetic contribution to disease risk. To do this, we will suggest some useful summary measures.

**Gini index**. First, we use the Lorenz curve, and its summary measure the Gini index. Although rarely used in medicine and epidemiology, this metric adequately describes the variation in disease risk[25,26]. Importantly, it allows for comparison across measurement scales; the Gini index does not depend on the cumulative risk of a disease in a population (or the total size of an economy), neither on the size of the population itself. It only relies on the relative mean absolute difference between individuals[26]. Crudely, the Gini index is a number between 0 and 1, describing the inequality in disease risk across individuals. More precisely, the Lorenz curve is represented by a function $L(S)$, in which $S$ is a cumulative proportion of the population, and $L(S)$ is the fraction of the total risk that is carried by $S$. E.g. if the risk is equal among subjects in the population, the fraction of risk carried by any 50% of the population would be $L(0.5) = 0.5$, which

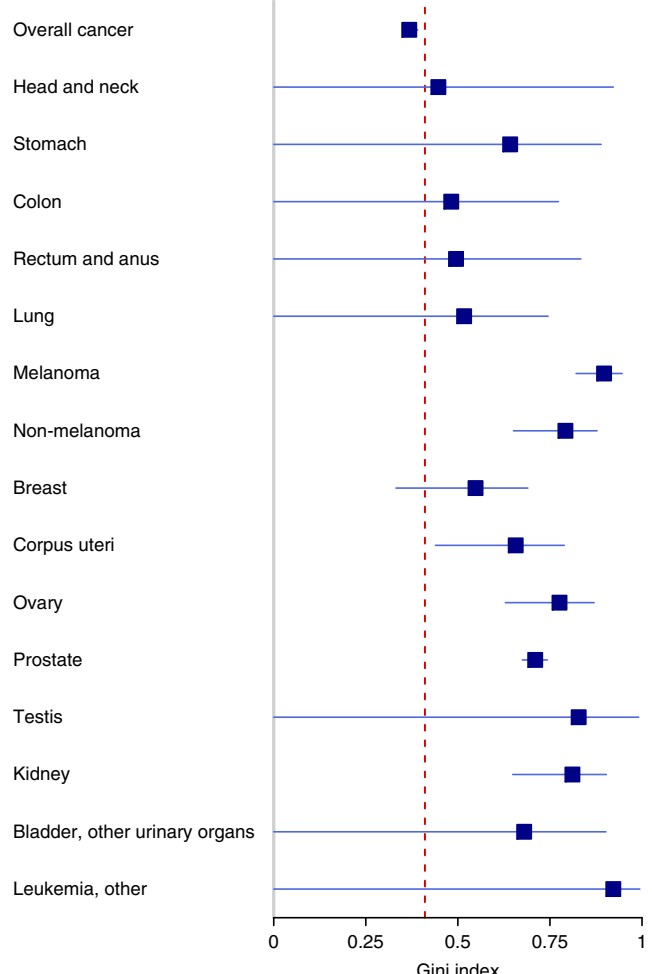

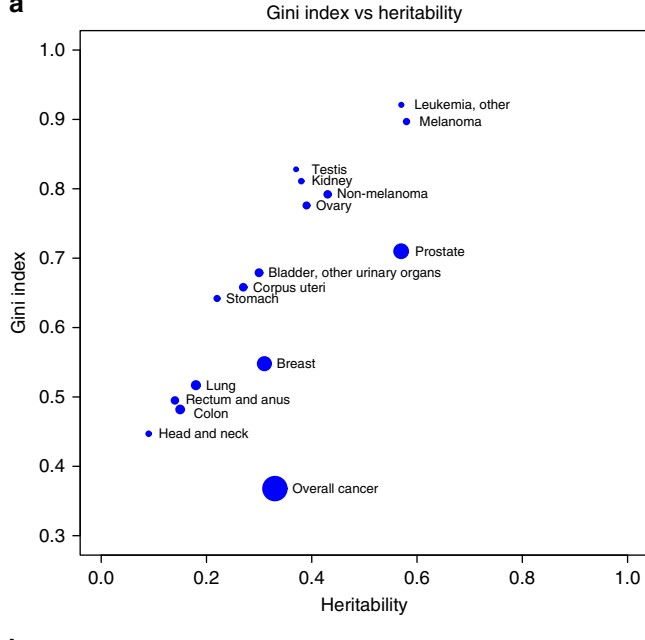

**Fig. 2** Gini indices for 15 common cancers. The Gini indices with 95% confidence intervals are derived by using data from Nordic twin registries[20]. The red dashed line marks the Gini index of income in the USA

means that the Lorenz curve is a straight line. The Gini index is a ratio describing the deviation from this straight line, which can be interpreted as a coefficient of deviation in risk, either on the absolute or the relative scale[26] (A formal mathematical derivation is found in the Methods section).

In our context, a Gini index of 0 means that everybody has the same genetic risk to a particular cancer, whereas a Gini index of 1 implies maximum inequality in risk across individuals. The Gini index is widely used in economics and demography, e.g., to study inequality in income and wealth. In Fig. 2, we show the Gini index for 15 common cancers. The Gini index is derived by using the heritability $h^2$ and life-time risk estimates form a recent Nordic twin study[20]. The red dashed line denotes the Gini index of income in the USA, using data from the World Bank[27]. Interestingly, the plot reveals a major inequality in cancer risk for the common cancers. For all specific cancers, the inequality in genetic risk seems to be larger than the inequality in income in the USA. We also studied the genetic risk of cancer overall, using the heritability of acquiring any type of cancer. This heritability estimate is lower than the individual cancers[20], which is expected because a factor increasing the risk of a particular cancer does not necessarily increase the risk of other cancers. Still, the Gini index of acquiring any type of cancer was almost as large as the Gini index for income in the USA.

We have displayed the relation between the Gini index and the heritability (Fig. 3a), and the relation between the Gini index and

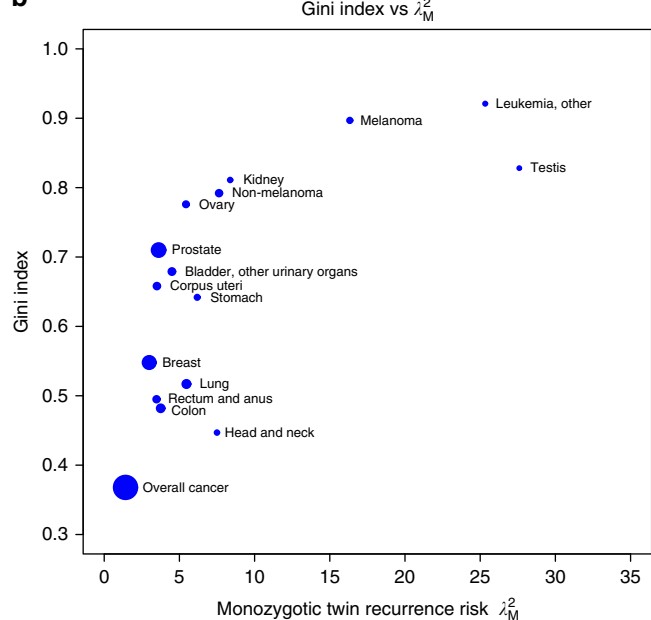

**Fig. 3** Gini indices, $h^2$ estimates and twin recurrence risks are not co-linear. **a** The relation between Gini indices and heritability estimates are displayed for 15 common cancers. **b** The relation between Gini indices and monozygotic twin recurrence risks ($\lambda_M$). The area of each circle is proportional to the life-time risk of the correspondin cancer

the observed relative risk in monozygotic co-twins of affected individuals ($\lambda_M$) (Fig. 3b). The areas of the circles are proportional to the life-time risk of the cancers. The three different measures of genetic contribution are related, but not co-linear, indicating that they capture non-overlapping information about the risk of disease. In particular, for cancer sites with similar heritability, the Gini index is relatively larger for the rarer sites.

**Quantile ratios**. Alternatively, we may study the inequality in risk by using a quantile ratio. The population is partitioned into subset according to quantiles of genetic risk, and we may estimate the ratio of affected individuals in the highest risk partition

**Table 1 Summary measures of the genetic risk of 15 common cancers**

| Site | Obtained from Mucci et al.[20] | | | | | | | |
|---|---|---|---|---|---|---|---|---|
| | $h^2$ (%) | $env^2$ (%) | AR (%) | $\lambda_M$ | $GC_{h^2}$ | $GC_{beta}$ | $RR_{20:20}$ | $RR_{interv}$ |
| Overall cancer | 33.0 | 0.0 | 32.4 | 1.4 | 0.37 | 0.37 | 9 | 0.64 |
| Head and neck | 9.0 | 26.0 | 0.8 | 7.5 | 0.45 | 0.84 | 12 | 0.55 |
| Stomach | 22.0 | 6.0 | 1.1 | 6.2 | 0.64 | 0.81 | 66 | 0.34 |
| Colon | 15.0 | 16.0 | 2.9 | 3.8 | 0.48 | 0.71 | 16 | 0.51 |
| Rectum and anus | 14.0 | 10.0 | 1.9 | 3.5 | 0.49 | 0.68 | 17 | 0.50 |
| Lung | 18.0 | 24.0 | 3.2 | 5.5 | 0.52 | 0.80 | 22 | 0.48 |
| Melanoma | 58.0 | 0.0 | 1.2 | 16.3 | 0.90 | 0.93 | 69,946 | 0.05 |
| Non-Melanoma | 43.0 | 0.0 | 1.9 | 7.6 | 0.79 | 0.85 | 997 | 0.17 |
| Breast | 31.0 | 16.0 | 9.4 | 3.0 | 0.55 | 0.66 | 36 | 0.45 |
| Corpus uteri | 27.0 | 0.0 | 2.0 | 3.5 | 0.66 | 0.69 | 87 | 0.33 |
| Ovary | 39.0 | 0.0 | 1.6 | 5.4 | 0.78 | 0.79 | 610 | 0.19 |
| Prostate | 57.0 | 0.0 | 10.5 | 3.6 | 0.71 | 0.73 | 689 | 0.26 |
| Testis | 37.0 | 24.0 | 0.5 | 27.6 | 0.83 | 0.96 | 1339 | 0.13 |
| Kidney | 38.0 | 0.0 | 0.8 | 8.4 | 0.81 | 0.85 | 1049 | 0.15 |
| Bladder, other urinary organs | 30.0 | 0.0 | 2.2 | 4.5 | 0.68 | 0.75 | 120 | 0.30 |
| Leukemia, other | 57.0 | 0.0 | 0.6 | 25.3 | 0.92 | 0.95 | 181350 | 0.03 |

Summary measures of genetic cancer risk are displayed. Here, $h^2$ denotes heritability of liability estimates, and $env^2$ denotes the contribution of common environmental factors to the variance of liability. AR denotes the absolute risk of cancer, and $\lambda_M$ denotes the recurrence risk in monozygotic co-twins. The four leftmost columns are obtained from Mucci et al.[20] $GC_{h2}$ denotes the Gini index derived from the heritability method. $GC_{beta}$ denotes the Gini index derived from the beta distribution. $RR_{20:20}$ denotes the ratio of mean risks from the upper vs the lower 20 percentile. $RR_{interv}$ describes the relative risk after an intervention in which those in the upper 20 percentile are manipulated to achieve the average risk in the lower 20 percentile

compared to the lowest risk partition. This metric is also frequently used to compare incomes in economics, e.g., the 20:20 ratio ($RR_{20:20}$) which assess the 20% richest compared to the 20% poorest of a population. Table 1 shows the $RR_{20:20}$ of genetic risk, which highlight a substantial difference in risk across subgroups; those in the highest 20 percentile carry substantially more of the disease burden than those in the lowest 20 percentile. In comparison, $RR_{20:20}$ for income is ~5 in the UK and ~9 in the USA[28].

**A hypothetical intervention**. Related to quantile ratios, we may estimate the effect of hypothetical interventions on particular risk groups. Suppose, for example, that we were able to reduce the genetic risk of each individual in the upper 20 percentile to the average risk in the lowest 20 percentile. This question could be relevant for public health professionals, because it suggests the potential benefit of identifying and subsequently intervening on high-risk populations.

We could calculate the relative risk of such interventions, assuming that the environment is left unaltered. Indeed, this relative risk is immediately obtained from the cumulative risk distribution. Let $y_{20}$ denote the 20 percentile of genetic risk and let $y_{80}$ denote the 80 percentile. Then

$$RR_{interv.} = \frac{\int_0^{y_{80}} y f_Y(y)\mathrm{d}y + \int_0^{y_{20}} y f_Y(y)\mathrm{d}y}{E(Y)}.$$

Relative risk estimates after such hypothetical interventions are found in Table 1. Indeed, these risk estimates also suggest a major contribution of genes to disease development; if we, e.g., were able to reduce the risk of prostate cancer in the upper 20 percentile to the average risk in the lower 20 percentile, we would reduce the number of cancers by a proportion of $1 - 0.26 = 0.74$.

**Using different sources of heritability data**. Heritability data may not only be derived from twin studies. Genome-wide association studies (GWAS) allows for the calculation of heritability estimates without relying on family structures[29,30]. These estimates account for the variability due to genetic variants tagged by single-nucleotide polymorphisms (SNPs), usually with a population frequency above 1–5%. Such array heritability

estimates are therefore considered to be lower bounds of the overall heritability, but may yield important information about the inequality in risk due to genetic variants associated with common SNPs. Lu et al.[29] estimated array heritability for a range of cancers, highlighting that array estimates captures approximately half the heritability from older twin studies. We may immediately apply our approaches to explore the inequality in cancer risk due to genetic variants tagged by SNPs. This could yield insight into, e.g., the benefit of targeting genetic variants tagged by SNPs in future interventions. In Fig. 4, we display the Gini indices derived from the array heritability estimates in Lu et al.[29], again highlighting the substantial inequailty in genetic risk.

**Alternative to the threshold model**. Although frequently used, the assumptions of the heritability of liability model are not necessarily satisfied[1]. Considering the liability to be normally distributed is convenient and may agree with the central limit theorem, but testing this assumption is usually infeasible in practice[7,24], and it may not be robust if the genetic risk is determined by few, rare genes[1]. When using twin data, we usually assume no gene-environment interaction on the liability scale[1,31], and we consider monozygotic- and dizygotic twins to share the same amount of environmental factors. Another issue is the confidence intervals of heritability and common environmental components, which are often wide even when hundreds of thousands are included in the study[20].

Until now we have based our results on the heritability of liability assumptions. We may, however, suggest a different approach that does not rely on the concept of heritability. We achieve this by assuming that the risk due to both heritable factors and common environment follows a parametric distribution. First, we let this distribution be the beta distribution, which allows for a wide range of shapes of the risk distribution and is bounded by 0 and 1. Importantly, in this model the risk distribution is uniquely defined by the observed recurrence risk (e.g., $\lambda_m$) and the disease prevalence[6]. First, we use the beta model to investigate the risk distribution due to the total effect of genes and shared environment. That is, this measure will capture the maximum inequality in risk due to genes and shared environment. Hence, we would generally assume that inequality

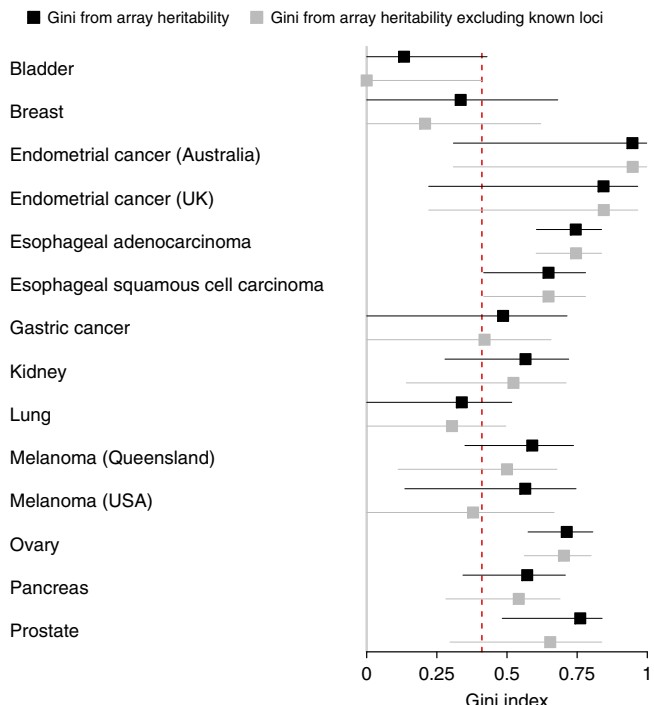

**Fig. 4** Gini indices derived from array heritability estimates. Gini indices with 95% confidence intervals are calculated from array heritability estimates derived from Lu et al.[29] The black boxes are based on array heritability removing loci with known association with the cancers. The red dashed line marks the Gini index of income in the USA

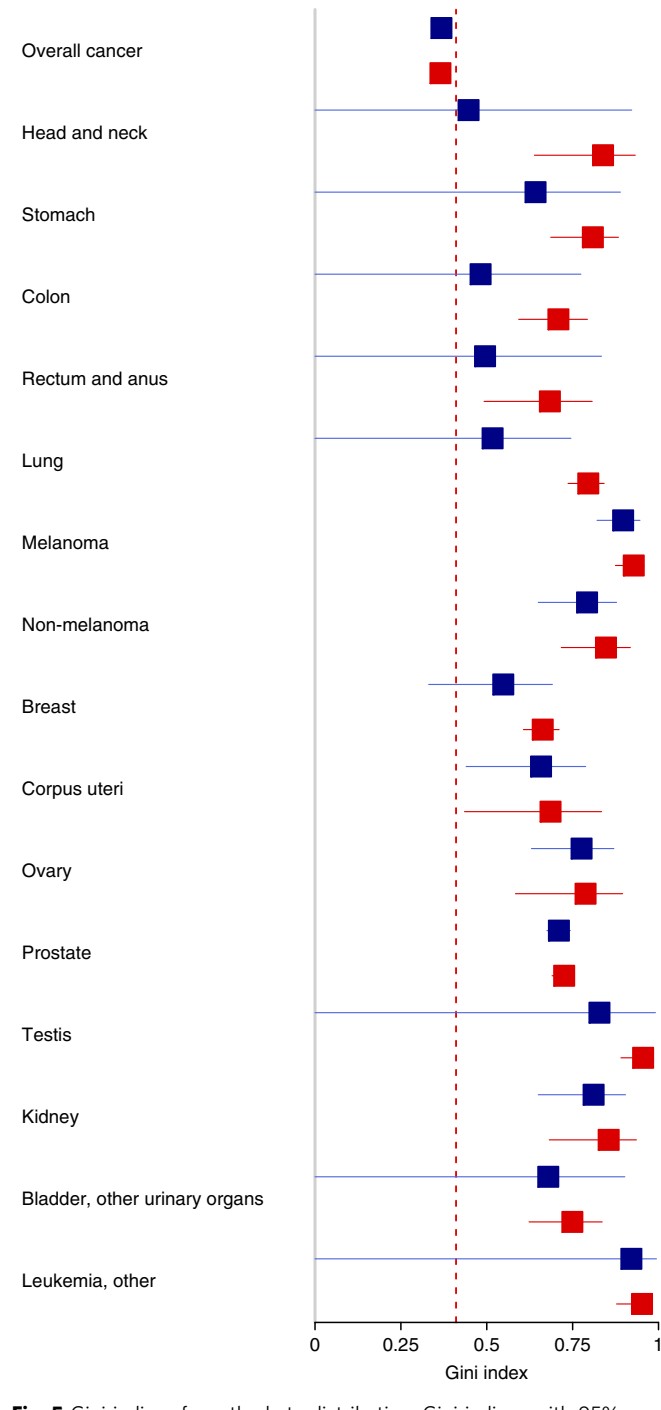

**Fig. 5** Gini indices from the beta distribution. Gini indices with 95% confidence intervals are displayed for the twin estimates in Fig. 2 (blue) together with estimates from the alternative beta distribution (red). The red dashed line marks the Gini index of income in the USA

measures from this approach, e.g., the Gini index, are larger in magnitude than the heritability based estimates. Intuitively, the differences should be relatively large if the shared environmental component is substantial, and relatively small if the common environmental component is minor. In Table 1, the Gini index from the beta models ($GC_{beta}$) are shown together with the Gini index from the heritability model ($GC_{h^2}$). The Gini indices from the beta model are generally larger than the estimates from the heritability model. As expected, the discrepancy is larger for the cancers with larger shared environmental components, which may be obtained by twin data as the fraction of the variance on the liability scale due to shared environment[20] ($env^2$ in Table 1). A plot similar to Fig. 2 including the beta Gini estimates is found in Fig. 5. For the cancers that were studied in both Mucci et al.[20] and Lu et al.[29], we have also compared twin heritability, array heritability and the estimates derived in this section (Fig. 6).

We may also use similar derivations for other distributions than the beta distribution. In particular, a distribution equal to $f_Y(y)$ in Eq. (4) of the Methods section could be derived directly by using estimates of $\lambda_M$ and the life-time disease risk. Then, we replace $h^2$ by $h^2_{env}$ in Eq. (4), and we let $h^2_{env}$ be a parameter that determines the shape of $f_Y(y)$. Indeed, we may interpret $h^2_{env}$ as the fraction of variance on the liability scale due to genes and common environment.

## Discussion

The contribution of heritable factors to major diseases is debated[14,16]. The antagonising views may arise due to ambiguous use of terminology and misinterpretation of model assumptions[3,18,19,32]. To gain deeper insight into the importance of genetic factors in cancer development, we have studied the absolute genetic risk distribution, under explicitly defined models. Thereby we can use measures of inequality that may be easier to understand than heritability itself,

e.g. the Gini index and the 20:20 ratio. These measures may be particularly desirable, because comparisons across scales can be made. Indeed, these measures are widely used in economics and demography, and they have also been successfully applied in biology previously[33].

Our results suggest that 15 common cancers show a major inequality in the genetic susceptibility to disease. As a curious comparison, we show that the inequality in cancer risk is larger than the income inequality in the USA. We must emphasise,

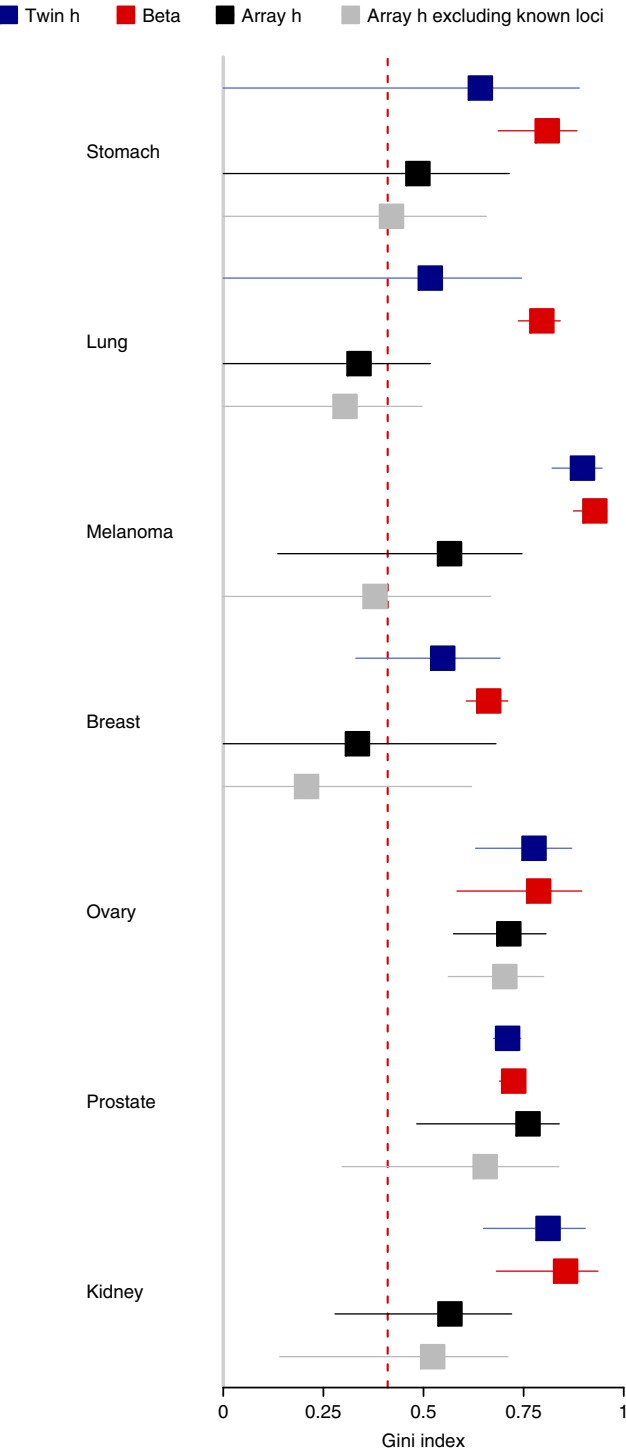

**Fig. 6** Comparing Gini indices from different risk distributions. The Gini indices with 95% confidence intervals displayed in Figs. 2, 4 and 5 are shown together. Only the cancers that were reported in both Mucci et al.[20] and Lu et al.[29] are included. The red dashed line marks the Gini index of income in the USA

least in theory. The information on risk inequality may be useful for public health professionals and other decision makers, when prioritising future prevention strategies and research projects. In particular, being able to identify high-risk individuals, and target these individuals for genetic or environmental interventions could be cost-effective strategies.

Fundamentally, our results put the debated role of chance in cancer development into perspective[8,34,35]: Irrespective of the definition of chance and the role of randomness in cancer development, we show that the genetic risk varies considerably across individuals. This points to major genetic variability in the individual risk of acquiring cancer. These findings do not contradict the results by either Tomasetti et al.[34,36] or Wu et al.[14] Rather, Tomasetti et al.[34] suggest that the cancer incidence at a site is strongly correlated with the number of baseline stem cell divisions at this site. Thereby they study heterogeneity between sites. We rather study heterogeneity within a cancer site, and suggest that environmental and genetic factors lead to major differences between individuals. Despite the seemingly random nature of stem cell mutations, there may be currently unknown processes, which vary across individuals, that influence the risk of particular cancers. Some individuals may be loaded with considerably higher risk than others, due to genetic or common environmental factors. We may denote these individuals as "unlucky". However, it is not necessarily sensible to assume that they are unlucky due to fundamentally random events[19].

## Methods

**Deriving the distribution of absolute risk**. We will show how the absolute genetic risk distribution is derived from the liability threshold model. To do this, we use the conventional assumptions of the liability model. Let the liability $L \sim N(\mu = 0, \sigma^2 = 1)$ be the sum of several components, and let $\Phi(z)$ denote the cumulative standard normal distribution. An individual is affected by disease $X$ with life-time risk $\Pr(X = 1) = 1 - q$ if

$$L \geq \Phi^{-1}(q).$$

To obtain estimates of $h^2$, it is usually assumed that $L$ has a genetic component

$$L_G \sim N(\mu = 0, \sigma^2 = h^2),$$

which is independent of the other components. We aim to find

$$\Pr(X = 1|\iota_G) = y.$$

We define $L_E = L - L_G$, which is the component of $L$ not determined by genotype. Usually, $L_G$ and $L_E$ are assumed to be independent, and therefore $L_E \sim N(0, 1 - h^2)$. Let $L_G = \iota_G$. Then,

$$L|\iota_G : N(\iota_G, 1 - h^2).$$

We are now able to express the probability of disease, given the genetic liability

$$
\begin{aligned}
g(\iota_G) &= P(X = 1|\iota_G) \\
&= P(L > \Phi^{-1}(q)|\iota_G) \\
&= P(L|\iota_G > \Phi^{-1}(q)) \\
&= \Phi\left(\frac{\iota_G - \Phi^{-1}(q)}{\sqrt{1 - h^2}}\right).
\end{aligned}
\tag{1}
$$

This relation has been graphically illustrated by Smith[21] and a mathematical expression was suggested by Mendell and Elston[22]. Due to the probit relation between $t_G$ and the absolute risk in Eq. (1), the liability threshold model has also been denoted a probit model[23].

We are interested in how $y$ varies among individuals in the population. Hence, we view $Y = g(L_G)$ as a random variable and let $g^{-1}(Y) = L_G$. Then

$$
\begin{aligned}
g(L_G) &= \Phi\left(\frac{L_G - \Phi^{-1}(q)}{\sqrt{1 - h^2}}\right) \\
g^{-1}(Y) &= \Phi^{-1}(Y)\sqrt{1 - h^2} + \Phi^{-1}(q)
\end{aligned}
\tag{2}
$$

**Simulating the distribution of Y**. Equation (1) allows us to simulate the distribution of $Y$ for a particular disease. To do this, we simply draw a standard Gaussian variable for each subject, which represents the genetic liability, and then

however, that our main results are based on the basic assumptions of the heritability estimates. In particular, we cannot immediately extrapolate the results outside the study populations.

Nevertheless, the major inequalities in risk suggest that many cancer cases are preventable in principle[18]. Even though preventative strategies are lacking today, our analysis therefore suggests that undiscovered targets for interventions may exist, at

transform this variable into an absolute risk. The procedure can be described more formally by the following algorithm:

1. Obtain $h^2$ and the population life-time prevalence $1 - q$ of the disease, e.g. from published data.

2. For each $i$ in $(1, \ldots, n)$, draw the individual liability $t_{G,i}$ from a normal distribution

$$L_{G,i} \sim N(\mu = 0, \sigma^2 = h^2)$$

3. For each $i$, calculate the genetic risk $y_i$ from Eq. (1)

$$y_i = \Phi\left(\frac{t_{G,i} - \Phi^{-1}(q)}{\sqrt{1 - h^2}}\right)$$

**Derivation of the distribution of Y**. We may also express the distribution of $Y$ algebraically. The probability density of $Y$ is expressed as

$$f_Y(y) = f_{L_G}\left(g^{-1}(y)\right) \times \frac{\mathrm{d}g^{-1}(y)}{\mathrm{d}y}$$
$$= f_{L_G}\left(g^{-1}(y)\right) \frac{1}{g'(g^{-1}(y))}, \tag{3}$$

where $f_{L_G}$ denotes the distribution function of $L_G \sim N(0, h^2)$. Furthermore

$$g'(g^{-1}(y)) = \frac{1}{\sqrt{2\pi}\sqrt{1-h^2}} \times e^{-\left(\frac{g^{-1}(y)-\Phi^{-1}(q)}{\sqrt{1-h^2}}\right)^2}{2}$$
$$= \frac{1}{\sqrt{2\pi}\sqrt{1-h^2}} \times e^{-\frac{(\Phi^{-1}(y))^2}{2}}.$$

Finally we plug into Eq. (3) to find

$$f_Y(y) = \frac{\sqrt{1-h^2}}{\sqrt{h^2}} e^{-\frac{g^{-1}(y)^2}{2h^2}} e^{\frac{(\Phi^{-1}(y))^2}{2}}$$
$$= \frac{\sqrt{1-h^2}}{\sqrt{h^2}} e^{-\frac{\left(\Phi^{-1}(y)\sqrt{1-h^2}+\Phi^{-1}(q)\right)^2}{2h^2} + \frac{(\Phi^{-1}(y))^2}{2}}. \tag{4}$$

By the definition of $Y$, we have that

$$E(Y) = E_{L_G}(P(X = 1 | t_G)) = P(X = 1) = 1 - q.$$

The variance of $Y$ can be found numerically by solving

$$\mathrm{VAR}(Y) = E(Y^2) - E(Y)^2 = \int_0^1 y^2 f_Y(y)\mathrm{d}y - (1-q)^2. \tag{5}$$

These derivations allow us to study how the absolute risk due to genetic differences is distributed in the population.

**Theoretic derivation of the Gini index**. We will present a formal definition of the Gini index as a function of the Lorenz curve. Let $f_Y$ and $F_Y$ be the probability density function (pdf) and cumulative density function (cdf) of $Y$, respectively. The Lorenz curve of the distribution of Y is defined as

$$L(x) = \frac{1}{E(Y)} \int_0^x t f_Y(t)\mathrm{d}t, \quad 0 \le x \le 1.$$

The Gini index of the distribution of $Y$ is then defined as

$$G_Y = 2\int_0^1 (F_Y - L(F_Y))\mathrm{d}F_Y$$
$$= 2\int_0^1 (F_Y(x) - L(x))f_Y(x)\mathrm{d}x.$$

The last equality (the integral limits) follows since $f_Y$ has support [0,1]. In general, the Gini index of the distribution of $Y$ may easily be found using numerical integration. For a Beta($\alpha, \beta$) distributed variable, the Gini index is explicitly given as

$$G_{\mathrm{Beta}} = \frac{2B(2\alpha, 2\beta)}{\alpha B(\alpha, \beta)^2},$$

where $B$ is the beta function[37].

**Risk due to heritable factors and shared family environment**. We assume that the risk of a particular cancer varies continuously across individuals in the population. More precisely, let $X_i$ be a binary variable taking value 1 if a subject is affected and 0 if a subject is unaffected. The probability of developing cancer in individual $i$, $p_i = P(X_i = 1)$, is drawn from a distribution $f(p_i)$ with support [0,1] and mean $\mu = E(p_i)$. Let $f(p_i)$ follow a parametric beta distribution, which allows for a

range of shapes. To completely specify $f(p_i)$, we must define $E(p_i)$ and $\mathrm{VAR}(p_i)$. We find $E(p_i)$ using published data on the life-time incidence of the disease $I_{\mathrm{life}}$. To derive an estimate of $\mathrm{VAR}(p_i)$, we make use of studies on monozygotic (MZ) twins. Following the terminology of Risch[3], let $\lambda_r$ denote the risk ratio of a relative of an affected individual. We assume that $p_i$ is equal in a pair of MZ twins. We interpret $p_i$ as the risk of disease due to heritable factors and shared family environment. Then we find $\lambda_M$, the risk ratio for disease given a co- MZ twin is affected

$$\lambda_M = \frac{P(X_2 = 1 | X_1 = 1)}{P(X_i = 1)}$$
$$= \frac{P(X_2 = 1, X_1 = 1)}{P(X_i = 1)P(X_1 = 1)}$$
$$= \frac{P(X_2 = 1, X_1 = 1)}{P(X_i = 1)^2}, \text{ since } P(X_1 = 1) = P(X_i = 1) \tag{6}$$
$$= \frac{E(p_i^2)}{E(p_i)^2}, \text{ since } P(X_1 = 1) = P(X_2 = 1)$$
$$= 1 + \frac{\mathrm{VAR}(p_i)}{E(p_i)^2}.$$

Using estimates of $\lambda_M$ from MZ twin studies, we can find

$$\mathrm{VAR}(p_i) = (\lambda_M - 1)E(p_i)^2$$
$$\approx (\hat{\lambda}_M - 1)I_{\mathrm{life}}^2. \tag{7}$$

Hence, under these assumptions we can completely specify the distribution of risk in the population, $f(p_i)$, if estimates of the cumulative incidence ($I_{\mathrm{life}}$) and the twin recurrence risk ($\lambda_M$) are available. We may interpret this as follows: Each subject obtains a risk (probability of developing disease) due to genetic factors and common environment. Then, this probability, combined with unmeasured individual factors and chance, determines whether the subject gets the disease.

Indeed, we can use exactly the same approach to specify the probit liability distribution in the main text. Then, we use Expression (4) as a parameterisation of the probit liability distribution, with parameters $E(y) = 1 - q$ and $h_{\mathrm{env}}^2$. Here, we have replaced $h^2$ by $h_{\mathrm{env}}^2$ in Eq. (5), because it no longer denotes heritability. Rather, $h_{\mathrm{env}}^2$ is the fraction of the trait variance on the liability scale due to both heritable factors and common environment. Mathematically, $h_{\mathrm{env}}^2$ is a shape parameter of the the probit liability distribution. Then, we combine Expressions (3) and (5) to

$$(\lambda_M - 1)E(p_i)^2 - E(Y^2) + E(Y)^2 = 0$$
$$(\lambda_M - 1)(1 - q)^2 - \int_0^1 y^2 f_Y(y)\mathrm{d}y + (1 - q)^2 = 0$$
$$(\lambda_M - 1)(1 - q)^2 - \int_0^1 y^2 \frac{\sqrt{1-h_{\mathrm{env}}^2}}{\sqrt{h_{\mathrm{env}}^2}} e^{-\frac{\left(\Phi^{-1}(y)\sqrt{1-h_{\mathrm{env}}^2}+\Phi^{-1}(q)\right)^2}{2h_{\mathrm{env}}^2} + \frac{(\Phi^{-1}(y))^2}{2}} \mathrm{d}y + (1 - q)^2 = 0. \tag{8}$$

Indeed, Expression (8) can be solved numerically to find $h_{\mathrm{env}}^2$.

**Numeric results**. To derive our numeric estimates, we have used the results from Tables 2 and 3 in Mucci et al.[20] and Table 2 in Lu et al.[29] Confidence intervals were obtain by inserting the confidence bounds reported in Mucci et al.[20] and Lu et al.[29] into our expressions for genetic risk. For the beta distribution, we used the confidence intervals in Table 2 in Mucci et al.[20] for recurrence risks in monozygotic twins. All our numeric results were obtained by two independent approaches, numeric integration of analytic expressions and simulations. Both approaches yielded the same results.

**Code availability**. The computer code for all the calculations was written in R version 3.3.2 using RStudio version 1.0.136. This computer code is available in Supplementary Data 1.

**Data availability**. We have solely used data that are readily available in previously published articles[20,29].

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

## Acknowledgements

This work was partially supported by the Norwegian Cancer Society, grant number 4493570, and the Nordic Cancer Union, grant number 186031. We thank Odd O. Aalen for his valuable comments to the manuscript.

## Author contributions

M.J.S. conceived the study. M.J.S. and M.V. performed and interpreted the data analysis. M.J.S. and M.V. drafted and critically revised the article. M.J.S. and M.V. approved the final version of the manuscript.

## Additional information

**Competing interests:** The authors declare no competing financial interests.

