## [Peer Review File · Nature Communications]

Reviewers' comments:

Reviewer #1 (Remarks to the Author):

The results of the analysis largely repeat what is already known in terms of contribution of genetic and environmental factors to cancer, and are based on a database which has been used in many previous projects. The additional contribution of this analysis is unclear.

The meaning of the Gini index on wealth is unclear – is it a proxy for environmental (i.e., non-genetic) determinants of cancer?

Some of the cancers included in the analysis have well-established (and strong) environmental risk factors: it would have been important to factor this knowledge into the proposed model.

Reviewer #2 (Remarks to the Author):

This is a well constructed paper and was a pleasure to read. I have no major concerns regarding the content. Some suggestions are given below.

The part comparing the Gini index for "genes" and "genes+common environment" was a useful extension to the basic method outlined for mapping heritability to Gini index. It would be of further interest to extend the approach to estimates of "SNP heritability". SNP heritability refers to the component of heritability that is tagged by common variants on genotyping arrays. Estimates of SNP heritability have been estimated across a range of cancers - see <https://www.ncbi.nlm.nih.gov/pubmed/24943595> and <https://www.ncbi.nlm.nih.gov/pubmed/24562164>. It would be simple for the authors to compute the Gini index for these.

I'd prefer to see the computer code in 4.1 put on the authors website (or the journal website or github or similar) rather than be available on request.

On figure 4 it looks like the beta result is lower than the hsq result which is presumably just a quirk of the plotting program.

typos

p2 vary -> varies

p3 form -> from

Reviewer #3 (Remarks to the Author):

This manuscript deals with the ongoing debate regarding the relative contribution of genetic versus other factors in the development of cancer. It strikes me as both important and timely, but could benefit from addressing the following items:

1. In general, I hesitate at the use of the word 'causal' in describing models fit to observational data. At the very least, the authors should justify their use of the term.
2. The y-axis of Figure 1 is not clearly defined. Density of what?
3. Can the authors clarify what they mean in saying that the Gini index allows for comparison across measurement scales?

4. The relationship between the Lorenz curve and the Gini index should be more explicitly defined.
5. Note the typo 'form' instead of 'from' toward the bottom of Page 3.
6. I admit that I was previously unfamiliar with the Gini index, but it strikes me as odd that the value for overall cancer is lower than those of the individual cancers evaluated. Does that suggest that the inequality is low for the individual cancers that were not presented? Given that the cancers shown are the most frequent, it is surprising to me that the Gini index for overall cancer would not be something of an average of the indices of the 15 common cancers.
7. The authors mention that heritability of liability models rely on potentially problematic assumptions. They do not, however, speak to why these assumptions may be problematic.
8. I would argue that the authors should be careful in their usage of the term 'preventable'. At least they should be wholly explicit that the interventions that would prevent disease do not yet exist.

Dear Reviewers,

Thank you for your positive and constructive feedback on the manuscript. After addressing your comments and suggestions, we believe that the article has been significantly improved. Please see the point-by-point reply below.

Reviewers' comments:

Reviewer #1 (Remarks to the Author):

The results of the analysis largely repeat what is already known in terms of contribution of genetic and environmental factors to cancer, and are based on a database which has been used in many previous projects. The additional contribution of this analysis is unclear.

We have suggested approaches to better understand the substantial inequality in genetic cancer risk. To the best of our knowledge, these are new ideas. Indeed, we use subject matter data from a well-known source, and this is intended: We aim to explore how our inequality measures yield additional insight compared to the standard heritability estimates.

The meaning of the Gini index on wealth is unclear – is it a proxy for environmental (i.e., non-genetic) determinants of cancer?

The Gini index is not a proxy for environmental factors. The Gini index is a well-established concept in economics, and it is used as a measure of inequality of wealth or in income in a population. However, in our context, Gini index rather describes the inequality in cancer risk due to genetic factors. The reference to the Gini index for income in the USA is made for illustrational purposes: We highlight that the inequality of income directly can be compared to the inequality of genetic risk, using e.g. the Gini index. The reference made to the Gini index for wealth was a typo, and have been corrected (to income).

Some of the cancers included in the analysis have well-established (and strong) environmental risk factors: it would have been important to factor this knowledge into the proposed model.

Indeed, there are well-established environmental components for many cancers. However, it is not our aim to model already known risk factors. We only use heritability data (whether from family studies or GWAS) to explore the substantial inequality in *genetic* cancer risk.

Reviewer #2 (Remarks to the Author):

This is a well constructed paper and was a pleasure to read. I have no major concerns regarding the content. Some suggestions are given below.

The part comparing the Gini index for "genes" and "genes+common environment" was a useful extension to the basic method outlined for mapping heritability to Gini index. It would be of further interest to extend the approach to estimates of "SNP heritability". SNP heritability refers to the component of heritability that is tagged by common variants on genotyping arrays. Estimates of SNP

heritability have been estimated across a range of cancers - see

<https://www.ncbi.nlm.nih.gov/pubmed/24943595>

<https://www.ncbi.nlm.nih.gov/pubmed/24562164>

It would be simple for the authors to compute the Gini index for these.

This is a useful suggestion. We have computed Gini indices from heritability estimates based on SNP data in the revised manuscript, using the data from Lu et al. Please see page 5, Section "Using different sources of heritability data".

I'd prefer to see the computer code in 4.1 put on the authors website (or the journal website or github or similar) rather than be available on request.

We entirely agree that the computer code should be readily available. We have included the R code as Supplementary material to the revised submission. If the journal rather prefer the R-code on the website, we have no objections to that.

On figure 4 it looks like the beta result is lower than the hsq result which is presumably just a quirk of the plotting program.

Indeed, this is could be handled in the plotting program (R). We have revised the this figure (previously Figure 4) and we have added some figures with similar layout. To better display the confidence limits, we have kept the boxes below each other. However, the boxes are now non-overlapping, which may look better.

typos

p2 vary -> varies

This is now corrected.

p3 form -> from

This is now corrected.

Reviewer #3 (Remarks to the Author):

This manuscript deals with the ongoing debate regarding the relative contribution of genetic versus other factors in the development of cancer. It strikes me as both important and timely, but could benefit from addressing the following items:

1. In general, I hesitate at the use of the word 'causal' in describing models fit to observational data. At the very least, the authors should justify their use of the term.

We agree that the word causal should be used with great care. We have removed the word causal from the abstract. Now the word causal only appears in the introduction, referring to the paper by Tenesa (ref 7) which uses the phrase "causal model" similar to us.

In the introduction we also state that "To interpret the heritability, we must make assumptions about the underlying causal structure, i.e. we must define a causal model (ref. 1,7)." In the following

sentence we stress that interpretation of heritability estimates are not easy. This is meant to indicate that making assumptions about the underlying causal structure is not trivial.

2. The y-axis of Figure 1 is not clearly defined. Density of what?

The y-axis displays the probability density (i.e., such that the area under the curve is 1) of the genetic risk of the cancers. This is now explicitly stated in Figure 1.

3. Can the authors clarify what they mean in saying that the Gini index allows for comparison across measurement scales?

We have clarified this in the revised manuscript on page 3. Briefly, the Gini index does not depend on the cumulative risk of a disease in a population (or the total size of economy, when inequality in wealth is measured), neither on the size of the population itself. It only relies on the relative mean absolute difference between individuals. This makes it possible to compare the inequality of different units.

4. The relationship between the Lorenz curve and the Gini index should be more explicitly defined.

We have elaborated on this in the revised manuscript. In particular, the Methods section now contains a paragraph describing the mathematical relation between the Lorenz curve and the Gini index.

5. Note the typo 'form' instead of 'from' toward the bottom of Page 3.

This typo is now corrected.

6. I admit that I was previously unfamiliar with the Gini index, but it strikes me as odd that the value for overall cancer is lower than those of the individual cancers evaluated. Does that suggest that the inequality is low for the individual cancers that were not presented? Given that the cancers shown are the most frequent, it is surprising to me that the Gini index for overall cancer would not be something of an average of the indices of the 15 common cancers.

We agree that this should have been better explained. The Gini index of overall cancer is not expected to be an average of the 15 common cancers, and we have now explicitly stated this on top of page 4. A strong familial clustering yields a larger value of the Gini index than a weak clustering. When looking at overall cancer, there is no distinction between cancer types, and cancers that are more or less related will be grouped. Heuristically, the effect of an allele increasing the risk of one cancer will be "diluted" when merging all cancers together. We would therefore expect that the Gini index (as well as the heritability of liability, as seen in ref 20) is lower for cancer overall than the particular cancers.

7. The authors mention that heritability of liability models rely on potentially problematic assumptions. They do not, however, speak to why these assumptions may be problematic.

We have discussed some of the assumptions of the heritability of liability model on page 5, and added some references to this section. A key assumption is the additive decomposition of the genetic and environmental contribution to the liability. That is, there are no assumed gene-environment interactions on the liability scale. This does, however, not imply that there is no interactions on the risk scale, since the risk is not a linear function of the liability, see ref. 1. Still, the assumption may not be plausible in real-life. Another assumption is the normally distributed heritability, which may not be the case if rare, strong genes determine the genetic risk.

In the revised manuscript we give references to elaborate discussion of the weaknesses of the heritability of liability model (e.g. ref. 7 and 31)

8. I would argue that the authors should be careful in their usage of the term 'preventable'. At least they should be wholly explicit that the interventions that would prevent disease do not yet exist.

We agree that such interventions do not yet exist. In the revised manuscript we highlight that "preventable in principle" does not necessarily mean preventable today (see the 3rd paragraph of the discussion in the revised manuscript).

REVIEWERS' COMMENTS:

Reviewer #3 (Remarks to the Author):

The authors adequately addressed my previous comments.